# The geographical pension gap: Understanding the causes of inequality in China's pension funds

**Songbiao Zhang, Xining Wang, Huajin Li, Huilin Wang** *

School of Business, Hunan University of Science and Technology, Xiangtan, China

* 1150141@hnust.edu.cn

**Data Availability Statement:** All relevant data are available within the paper, its Supporting Information files, and on Havard Dataverse: https://doi.org/10.7910/DVN/3AHRDV.

## Abstract

The sustainability of social pension insurance is of great significance in guaranteeing the essential life of the elderly and promoting social stability. Based on the provincial panel data from 2012 to 2020, this study uses non-spatial measurement methods, ArcGIS visualization research methods, and geographic detectors to study the regional differences in China's pension fund balances and the underlying influencing factors. Compared with the traditional way of establishing regression equations to explore the correlation of influencing factors, geographic detectors can quantify the strength of each influencing factor and detect the interaction of different influencing factors. This study found that: First, the growth rate of China's overall pension fund balances has been declining yearly, with the fastest decline in northeast China, the middle in the Western and Central regions of China, and the slowest decline in Eastern China. Second, the spatial distribution of pension fund balances shows agglomeration characteristics, with high-value areas mainly distributed in Eastern China and low-value regions distributed primarily in Western and Northeastern China. Third, the overall Theil index for pension fund balances is trending down, but the Theil index for the Eastern region is on the rise. Fourth, seven factors, including the working-age population, the population aged 65 and above, and regional GDP, are the main factors that lead to regional differences in the balance of urban and rural residential insurance funds. Finally, the superimposed effects of each element are reflected in double-factor enhancement or non-linear enhancement relation.

## Introduction

Since implementing economic reforms and opening up, China has experienced rapid economic development and substantially reduced absolute poverty. However, the issue of relative poverty remains severe, and the widening income gap poses a significant obstacle to achieving common prosperity [1]. Establishing the old-age insurance system has been a crucial policy tool to narrow this income gap. Nevertheless, China faces the unique challenge of "getting old before getting rich" due to the relatively short period during which the old-age security system has been in place and the accelerated population aging, particularly among certain demographic groups [2].

**Funding:** This study was supported by the National Science Foundation of China (Project No. 42101172, funding receiver: Songbiao Zhang) and the Hunan Provincial Philosophy and Social Science Foundation (Project No. 22YBA133, funding receiver: Songbiao Zhang). The funders had no role in study design, data collection and analysis, decision to publish, or preparation of the manuscript.

**Competing interests:** The authors have declared that no competing interests exist.

Initially, China's pension system was designed to safeguard the well-being of older adults, prevent elderly poverty, and reduce income disparities among older individuals. Currently, the pension system in China has achieved nearly universal coverage among the population, with an increasing participation rate. However, due to inadequate overall planning of the pension insurance fund at the national level, a significant disparity exists in the income and expenditure of pension funds across different provinces. As a result, some provinces face shortfalls in pension fund revenue, resulting in a decline in the growth rate of the fund balance. Consequently, this not only negatively impacts efforts to alleviate poverty among older adults but also exacerbates regional income disparities among this age group, contradicting the original intent of the pension system [3]. Furthermore, demographic changes are generating global concerns regarding the long-term financial sustainability of pension schemes [4–6], a matter of particular urgency for China. Moreover, as the pension system covers the most significant number of people in China, the economic sustainability of pension funds is also facing the same challenges [7]. The major problem is that due to the uneven regional development in China, there are significant differences in the financial sustainability of pension funds in various regions [8]. For example, according to data from the National Bureau of Statistics, the province with the highest pension fund balance in 2020 is 38 times that of the lowest [9]. This huge regional difference hurts the sustainability of pension funds, but also to a certain extent. It weakens the fairness of the pension system and violates the original intention of the design of the pension system to take into account both efficiency and right [10].

The reason is that, on the one hand, areas with low pension fund balances tend to have relatively high levels of old-age support. As the population's average life expectancy continues to prolong, rapid aging has led to a substantial increase in expenditure on pension funds. At the same time, the working-age population in areas with small pension fund balances is relatively tiny, and pension fund income will continue to decline, bringing more significant pressure on government finances; on the other hand, in areas with small pension fund balances, the pension The level of benefits received is also relatively low. As a result, pension funds are an essential source of income for rural elderly residents [11], and significant regional differences in pension fund balances will widen the income gap among older adults in different regions [12]. Furthermore, due to the interaction between pension fund balance and population flow [13], the smaller the pension fund balance and the lower the pension level, the more the outflow of the labor force will be exacerbated [14]. The loss of labor force will reduce the pension supply of the place of emigration, which will not only aggravate the financial crisis of the pension fund of the area of emigration but also lead to the further expansion of the level of economic development between regions [15]. Therefore, it is of great theoretical and practical significance to understand the time-progressing process of the regional gap in pension fund balances, the evolution of the spatial distribution, and to find out the factors that cause and propose relevant solutions accordingly.

This study explores the regional differences in pension fund balances in China and the reasons behind them, aiming to explore the following questions: (1)To explain the evolution of China's pension fund balances from a temporal perspective; (2)To explain the evolution of China's pension balances from a spatial perspective; (3)To detect the strength of the influencing factors that cause the regional gap in pension funds and the interaction of each factor. This study's contributions include presenting the evolution and distribution of regional gaps in China's pension fund balances from multiple dimensions of time and space, fully reflecting the time course and distribution of regional gaps in China's pension fund balances. Secondly, incorporate the economic, demographic, fiscal, and institutional aspects into a unified analysis framework, and quantify the strength of each factor's effect on the balance of regional pension funds without considering multicollinearity. Thirdly, the interdisciplinary method is used to

bring economic factor variables into the geographic model, effectively identify the interaction relationship between multiple factors, and provide a new way of analyzing the reasons for the regional gap of a particular economic factor.

This study focuses on the reasons for the regional disparity in China's pension fund balance. The research results show that the sustainability of pension fund balances is decreasing year by year, the regional gap in fund balances is increasing yearly, and the areas where funds are based are primarily concentrated in the developed Eastern coastal regions. Besides, the regional differences in pension fund balances are more affected by demographic and institutional factors. The interaction of each influencing factor shows a linear enhancement, indicating that formulating government policies must comprehensively consider various factors to achieve optimal results.

## Literature review

### Pensions

There are evident variations in pension systems among different countries and within other groups of people. For example, in the case of China, notable disparities exist in terms of urban-rural gaps, gender gaps [16], and regional gaps [17] within the pension system. Regional disparities manifest in several ways, including differences in coverage rates, pension treatment levels, pension insurance support rates, and pension balances.

China's pension system underwent a merger in 2014, combining the urban residents' endowment insurance and the new rural residents' basic endowment insurance. The system primarily targets individuals aged 16 and above [excluding school students] not covered by the employee's endowment insurance system. It mainly addresses low-income groups in the country, such as migrant workers and rural residents. Studies have shown that urban and rural pension income is a vital income source for the elderly population in rural areas, playing a significant role in preventing elderly poverty and maintaining social equity [18]. However, as a fund pool for regional pension distribution, pensions exhibit regional disparities in absolute amounts and growth rates. This puts additional pressure on the pension system's sustainability and can widen the income gap among older adults [12].

### Sustainability of pension insurance funds and influencing factors

Fiscal sustainability is the core principle of social security [19]. Existing international research provides a detailed analysis of pension fund income, expenditure, and future forecasts. Internationally, the safety of national pension funds is judged mainly by constructing actuarial models [20–22]. Most predictions for the financial sustainability of China's pension funds are made by constructing linear equations [23, 24]. There are many research results on the factors affecting the sustainability of pension funds, which can be summarized as fund investment management, population structure changes, and institutional settings. The investment management methods of pension funds in various countries are different. The effective allocation of pension funds can improve the level of pension investment income, realize the maintenance and appreciation of pension funds, and resolve the pension payment risks brought about by the aging population [25]. However, the inhibiting effect of population aging on the sustainability of pension funds is more evident in economically backward areas [26]. In addition, the definition of government responsibility by the pension system significantly impacts the sustainability of pension funds [27]. To deal with the financial unsustainability of pension funds under an aging population, scholars have put forward a series of policy recommendations, such as phased retirement [28], extending the retirement age [29], reducing contribution rates [30], and encouraging births and other reforms [31].

### Regional gap of pension funds and influencing factors

Research on the influencing factors of regional differences in pension funds, mainly focusing on systems, finance, economy, and population. For example, from an institutional point of view, a regional division of pooling is the main reason for regional differences in the balance of pension funds. China's future pension reform should speed up national pooling [24]. From the perspective of differences in fund responsibility subjects, there are differences in financial subsidies, collective subsidies, and pension fund investments in different regions [32], which also affect the regional differences in pension fund replacement rates to a certain extent [33]. Therefore, further balancing the burden of pension responsibilities between areas is necessary [34, 35]. From the perspective of unbalanced economic development, there is a significant gap in the level of economic development in the different regions. Therefore, economic factors such as regional GDP substantially impact regional differences in pension funds [36, 37]. From the perspective of changes in population structure, the provinces with net population inflows have less pressure on pension expenditures. In contrast, the regions with net population outflows have a more significant strain on pension expenditures, which has led to substantial differences in pension fund balance areas. Increasing the birth rate and optimizing the population structure are essential to enhance the sustainability of pension funds in various regions [38, 39].

Existing research on pension funds mainly focuses on the actuarial balance of the fund to pursue sustainability of the pension fund. However, the lack of fairness in the pension system brought about by the regional gap in pension funds is rarely involved. However, the research on the influencing factors of pension fund sustainability and regional disparity mainly focuses on a single element, needs to put all the influencing factors into a unified analysis framework, and consider the interaction of different factors.

## Methods

### Research object

This study takes 31 provinces in mainland China (excluding Hong Kong, Macao, and Taiwan) as the research object. According to the National Bureau of Statistics division, this study divides the 31 provinces into four parts: Eastern, Central, Western, and Northeastern.

### Data sources

Considering the data availability, this study's time range is limited to 2012–2020. The accumulated balance of pension funds in each province, the number of insured persons, per capita disposable income of residents, economic scale (GDP), fiscal subsidies, social security expenditures, and other indicators involved in this article come from the "China Statistical Yearbook," "China Labor Statistical Yearbook," "China Civil Affairs Statistical Yearbook," and statistical yearbooks of 31 provinces (municipalities, autonomous regions).

### Research methods

In measuring regional disparities, numerous calculation methods have been utilized in previous studies, including the coefficient of variation, Theil index, standard deviation, and Gini coefficient. However, when evaluating changes in the absolute gap between pension regions, comparing the absolute gaps at different time points can be misleading due to variations in the balance growth rate and the overall balance level. To address this issue, the relative gap is employed in this study, which eliminates the direct impact of the total size and enables a more

meaningful assessment of changes in regional disparities over time. In addition, the growth rate of fund balances in each region is calculated to reflect these changes.

Furthermore, China has historically exhibited significant disparities in fund balances among its Eastern, Central, Western, and Northeastern regions, primarily due to their different geographical locations. The advantage of utilizing the Theil index lies in its ability to measure and compute gaps between and within regions, making it a suitable choice for quantifying the discrepancies in fund balances.

Lastly, this study adopts the geographic detector method to identify multiple influencing factors contributing to regional fund balance gaps. Unlike conventional spatial regression analysis, the geographic detector method does not rely on linear assumptions or conditional restrictions. It offers an objective means of determining the degree of interpretation of independent variables on the dependent variable and provides the added advantage of detecting interactions between these factors.

## Pension fund balance growth rate

The accumulative balance growth rate of the pension fund is the growth of a particular year's accumulative fund balance relative to the previous year's. This indicator can reflect the sustainability of pension fund growth and fund revenue and expenditure changes. Therefore, using this indicator to examine regional differences in pension fund balances is significant. The specific calculation formula is as follows:

$$I = (R_t - R_{t-1})/R_{t-1} \tag{1}$$

In Formula (1), $I$ represent the growth rate of the accumulated balance of the pension fund; $R_t$ is the accumulated balance of the fund in year $t$; $R_{t-1}$ is the accumulated balance of the fund in year $t-1$.

## Theil index

This study selects the Theil index as the measurement index to study the overall difference in pension fund balances and the differences between regions and within regions. Further, it analyzes the contribution of disparities between and within groups to the general differences. Theil index and its decomposition formula are as follows [40]:

$$T = T_b + T_w \tag{2}$$

$$T = \sum_j \sum_i (\frac{Y_{ij}}{Y}) ln(\frac{Y_{ij}}{Y} / \frac{N_{ij}}{N}) \tag{3}$$

$$T_w = \sum_i (\frac{Y_i}{Y}) T_{wi} = \sum_i \sum_j (\frac{Y_i}{Y})\left(\frac{Y_{ij}}{Y}\right) ln(\frac{Y_{ij}}{Y} / \frac{N_{ij}}{N}) \tag{4}$$

$$T_b = \sum_i (\frac{Y_i}{Y}) ln(\frac{Y_i}{Y} / \frac{N_i}{N}) \tag{5}$$

$$T_{wi} = \sum_j (\frac{Y_{ij}}{Y_i}) ln(\frac{Y_{ij}}{Y_i} / \frac{N_{ij}}{N_i}) \tag{6}$$

Among them, $T$ is the overall Theil index, $T_b$ and $T_w$ are the differences between groups and within groups, respectively, and $T_{wi}$ is the Theil index in each region. In each formula, $i$ represents the four geographical regions of Eastern, Central, Western, and northeast China, respectively; $j$ represents each province in the geographical region; $Y$, $Y_i$, and $Y_{ij}$ represent the pension fund balances of the whole country, a region, and province, respectively; $N$, $N_i$, and $N_{ij}$ represent the number of provinces in the entire country, a region, and a province, respectively. To analyze the between-group contribution rate ($W_b$) and within-group contribution rate ($W_w$), this study defines the between-group contribution rate ($W_b$) as the contribution degree of the difference between groups to the overall difference. The within-group contribution rate ($W_w$) is defined as the contribution degree of the internal difference to the overall contrast; the calculation formula of the two is:

$$W_b = T_b/T \tag{7}$$

$$W_w = T_w/T \tag{8}$$

## Geographic detectors

The geographical detector method can detect the different influences of multiple factors in other spatial units and their interrelationships, including risk detection, factor detection, ecological detection, and interaction detection [41]. This study uses factor detection to measure the degree of explanation of factors affecting regional differences in pension fund balances and the changing trend of the strength of each element in different periods. Using interaction detection to determine whether the interaction of two parts and the impact of a single component on the regional differences in pension funds are stronger or weaker. The detection force value can be expressed as:

$$q_{D,H} = 1 - \frac{\sum_{h=1}^{L} n_h \sigma_h^2}{n\sigma^2} \tag{9}$$

In Formula (9), $q_{D,H}$ is the detection power value of the impact factor on the regional differences of pension funds; $n_h$ is the number of sample units in the sub-level region; $n$ is the number of sample units in the entire region; $L$ is the number of sub-regions; $\sigma^2$ is the variance of the sample size; $n_h \sigma_h^2$ is the variance of the sub-level area. The value range of $q_{D,H}$ is (0, 1). The larger the value, the greater the impact of the factor on the regional differences in pension fund balances; otherwise, the smaller it is. The principle of interactive detection is: if $P(X1 \cap X2) = P(X1)+P(X2)$, it means that the factors $X1$ and $X2$ are independent of each other; if Min $(P(X1), P(X2)) < P(X1 \cap X2) < $Max$(P(X1), P(X2))$, indicating that the single-factor nonlinearity weakens after the interaction between factors $X1$ and $X2$; if $P(X1 \cap X2) < $Min$(P(X1), P(X2))$, It means that the nonlinearity weakens after the interaction between $X1$ and $X2$; if $P(X1 \cap X2) > P(X1)+P(X2)$, it means that the nonlinearity increases after the interaction between $X1$ and $X2$. If $P(X1 \cap X2) > $Max$(P(X1),P(X2))$ and $P(X1 \cap X2) < P(X1)+P(X2)$, it means that the interaction between $X1$ and $X2$ is enhanced by two factors.

## Results

### Regional differences in pension fund balances

**Regional differences in the growth rate of pension fund balances.** From 2012 to 2020, the regional differences in the growth rate of pension fund balances are pronounced. The growth rate varies from province to province, but all regions show a clear downward trend (see Table 1). From the perspective of average growth rate, the Central and Western regions

are the fastest, reaching 24.42% and 23.34%, respectively; the northeast and Eastern regions are relatively slow, respectively, at 17.93% and 17.44%. It should be noted that the average growth rate is related to the base size of the pension fund balance. Taking Jiangsu Province and Jilin Province as examples, the pension fund balances of Jiangsu and Jilin Provinces were 68.98 billion yuan and 7.25 billion yuan in 2019, respectively, and 78.83 billion yuan and 8.57 billion yuan in 2020, with growth rates of 14.28% and 18.21% respectively. Due to the large base of the pension fund balance in Jiangsu Province, although the fund balance has increased by 9.85 billion yuan, the growth rate still needs to be higher. On the other hand, in Jilin Province, since the base of the fund balance itself is relatively small, although the fund balance has only increased by 1.32 billion yuan, the growth rate will be somewhat higher.

From the perspective of the decline in growth rate, the northeast region has the fastest decline, from 43.44% in 2012 to 14.99%, a drop of 28.45%. The Western and Central regions followed, with declines of 19.02% and 18.53%, respectively. The Eastern region declined the slowest, down only 12.04%. This is likely related to the population flow in the various areas, with regions with large net population inflows declining more slowly. In contrast, regions with sizeable net population outflows experienced faster declines. Taking the Northeast region as an example, since 2013, the permanent population of the three Northeastern provinces has seen a net outflow for seven consecutive years, with a cumulative discharge of 1.64 million people. The most outflow population is the working-age population, which directly leads to an increase in the dependency ratio of the elderly population, an increase in the system support rate, an increase in pension fund expenditure, and a decrease in income. Therefore, the financial sustainability of future funds in Northeast China is more prominent. This shows that although the fund balances in various regions are still growing, the growth rate has slowed. There are also significant differences in the growth rate of fund balances in multiple regions.

## Spatial distribution of pension fund balances

To reflect the regional differences more intuitively in pension fund balances, this paper selects the pension fund balance data for four years, 2012, 2015, 2018, and 2020, uses ArcGIS 18 for spatial visualization, and uses the natural breakpoint method to divide the four regions. It is divided into five grades: high-value area, sub-high value area, medium-value area, sub-low value area, and low-value area. Finally, the evolution picture of its spatial pattern is described. The results are shown in Fig 1.

Overall, the spatial distribution of pension fund balances presents prominent agglomeration characteristics, with different aggregation distributions in the Eastern, Central, Western, and Northeastern regions, and shows specific changes over time.

In 2012, the high-value areas were all distributed in the Eastern region. Half of the sub-high value areas are distributed in the Eastern region, and one province in the Central and Western regions has entered the sub-high value area. The median area is mainly distributed in the Central region. Sub-low and low-value areas are primarily distributed in the northeast and Western regions.

In 2015, three provinces in the high-value area were still distributed in the Eastern region. Still, Sichuan in the Western region and Henan in the Central region were also promoted to the high-value area. Sub-high value areas increased significantly, with 3, 3, and 2 provinces in the Eastern, Central, and Western regions entering the sub-high value area. The median area is relatively scattered, with 2, 2, and 3 provinces entering the median areas in the Eastern, Central, and Western regions. Sub-low and low-value areas are still distributed in the northeast and Western regions.

In 2018, only Shandong Province in the Eastern region remained among the provinces in the high-value area. There are only five provinces in the second-highest value area, one from

**Table 1. Growth rate of pension fund balances in various provinces in China.**

| Regions | Province | 2013 | 2014 | 2015 | 2016 | 2017 | 2018 | 2019 | 2020 | Average |
|---|---|---|---|---|---|---|---|---|---|---|
| **Eastern** | Beijing | 14.32% | 15.19% | 9.16% | 9.02% | 5.97% | 5.77% | 6.23% | 2.72% | 8.55% |
| | Shanghai | -0.41% | 3.06% | -0.27% | 4.46% | -1.03% | 6.54% | -1.23% | 11.06% | 2.77% |
| | Zhejiang | 14.80% | 7.64% | 3.52% | 4.57% | 0.66% | 3.29% | -1.53% | 62.14% | 11.89% |
| | Guangdong | 34.33% | 35.63% | 20.02% | 7.81% | 4.54% | 3.48% | 9.70% | 3.94% | 14.93% |
| | Fujian | 38.31% | 29.26% | 24.46% | 20.53% | 16.14% | 15.01% | 18.13% | 18.16% | 22.50% |
| | Hebei | 42.77% | 27.10% | 24.75% | 17.99% | 16.81% | 16.59% | 20.42% | 18.67% | 23.14% |
| | Tianjin | 45.55% | 20.02% | 15.35% | 37.14% | 20.94% | 8.06% | 5.83% | 6.08% | 19.87% |
| | Hainan | 53.54% | 34.21% | 71.08% | 46.13% | 27.84% | 26.53% | 23.15% | 11.71% | 36.77% |
| | Shandong | 25.10% | 31.22% | 21.02% | 20.31% | 20.32% | 19.85% | 14.17% | 14.68% | 20.83% |
| | Jiangsu | 15.47% | 11.64% | 16.14% | 14.32% | 12.11% | 12.82% | 8.12% | 14.28% | 13.11% |
| | Average | 28.38% | 21.50% | 20.52% | 18.23% | 12.43% | 11.79% | 10.30% | 16.34% | 17.44% |
| **Central** | Henan | 46.45% | 28.19% | 20.80% | 19.33% | 15.00% | 17.90% | 16.85% | 15.39% | 22.49% |
| | Jiangxi | -19.47% | 125.94% | 22.30% | 25.16% | 27.31% | 25.51% | 15.45% | 20.57% | 30.35% |
| | Shanxi | 22.45% | 29.26% | 23.34% | 19.33% | 20.03% | 15.15% | 15.73% | 15.30% | 20.07% |
| | Anhui | 57.16% | 33.06% | 24.84% | 21.49% | 20.04% | 23.72% | 21.06% | 22.17% | 27.94% |
| | Hunan | 62.56% | 29.72% | 28.20% | 20.48% | 22.32% | 14.82% | 16.79% | 12.97% | 25.98% |
| | Hubei | 47.53% | 25.91% | 22.20% | 21.91% | 23.02% | 22.98% | 22.32% | 19.10% | 25.62% |
| | Average | 36.11% | 45.35% | 23.61% | 21.28% | 21.29% | 20.01% | 18.03% | 17.58% | 24.42% |
| **Western** | Guangxi | -85.71% | 1196.08% | 34.49% | 24.30% | 25.61% | 14.63% | 19.67% | 22.95% | 156.50% |
| | Chongqing | 65.19% | 33.33% | 5.49% | 7.44% | 15.63% | 13.52% | 15.98% | 11.57% | 21.02% |
| | Inner Mongolia | 52.49% | 7.92% | 7.34% | 11.89% | 17.40% | 6.11% | 8.00% | 22.61% | 16.72% |
| | Ningxia | 47.06% | 28.80% | 19.88% | 21.24% | 16.67% | 18.68% | 14.51% | 15.90% | 22.84% |
| | Shaanxi | 40.62% | 34.62% | 21.62% | 15.22% | 12.92% | 15.22% | 15.45% | 16.77% | 21.55% |
| | Yunan | 93.84% | 32.10% | 25.98% | 21.96% | 20.98% | 13.03% | 12.10% | 65.92% | 35.74% |
| | Sichuan | 33.60% | 18.38% | 18.78% | 16.11% | 25.70% | 10.45% | 8.74% | 19.09% | 18.85% |
| | Qinghai | 34.83% | 37.50% | 30.91% | 24.07% | 23.51% | 18.73% | 19.34% | 27.29% | 27.02% |
| | Guizhou | 56.58% | 35.91% | 26.76% | 20.58% | 20.68% | 12.51% | 9.27% | 13.57% | 24.48% |
| | Xinjiang | 36.14% | 25.66% | 22.54% | 19.54% | 19.07% | 18.98% | 20.93% | 18.90% | 22.72% |
| | Tibet | 131.03% | 35.82% | 26.37% | 26.09% | 53.10% | 13.96% | 14.23% | 17.30% | 39.74% |
| | Gansu | 3.31% | 47.55% | 22.63% | 19.08% | 21.37% | 20.35% | 16.25% | 28.93% | 22.43% |
| | Average | 42.42% | 127.81% | 21.90% | 18.96% | 22.72% | 14.68% | 14.54% | 23.40% | 23.34% |
| **Northeast** | Heilongjiang | 46.51% | 14.29% | 7.54% | -3.14% | 33.33% | 15.57% | 23.24% | 16.65% | 19.25% |
| | Liaoning | 37.70% | 20.24% | 13.47% | 9.60% | 10.83% | 6.03% | 8.54% | 10.11% | 14.56% |
| | Jilin | 46.12% | 17.94% | 12.96% | 8.23% | 26.27% | 13.69% | 16.37% | 18.21% | 19.97% |
| | Average | 43.44% | 17.49% | 11.32% | 4.90% | 23.48% | 11.76% | 16.05% | 14.99% | 17.93% |

**Note:** Due to the highest extreme growth rate of pension fund balances in Guangxi, Tibet, and Jiangxi, they were excluded when calculating the average value.

the Western region, two from the Central region, and two from the Eastern region. The median area is mainly distributed in two contiguous areas and one province, namely Hebei and Tianjin contiguous areas, Shaanxi, Hubei, Hunan, and Jiangxi contiguous areas, and Yunnan province. There have been significant changes in the sub-low value area. In addition to the Western region, one province in the Central region has entered, and three provinces in the Eastern region have also entered. The low-value areas are still primarily distributed in the northeast and Western regions.

In 2020, only Shandong Province remained in the high-value area. Based on 2018, the sub-high value areas have squeezed into Hebei Province in the Eastern region, Hubei Province in

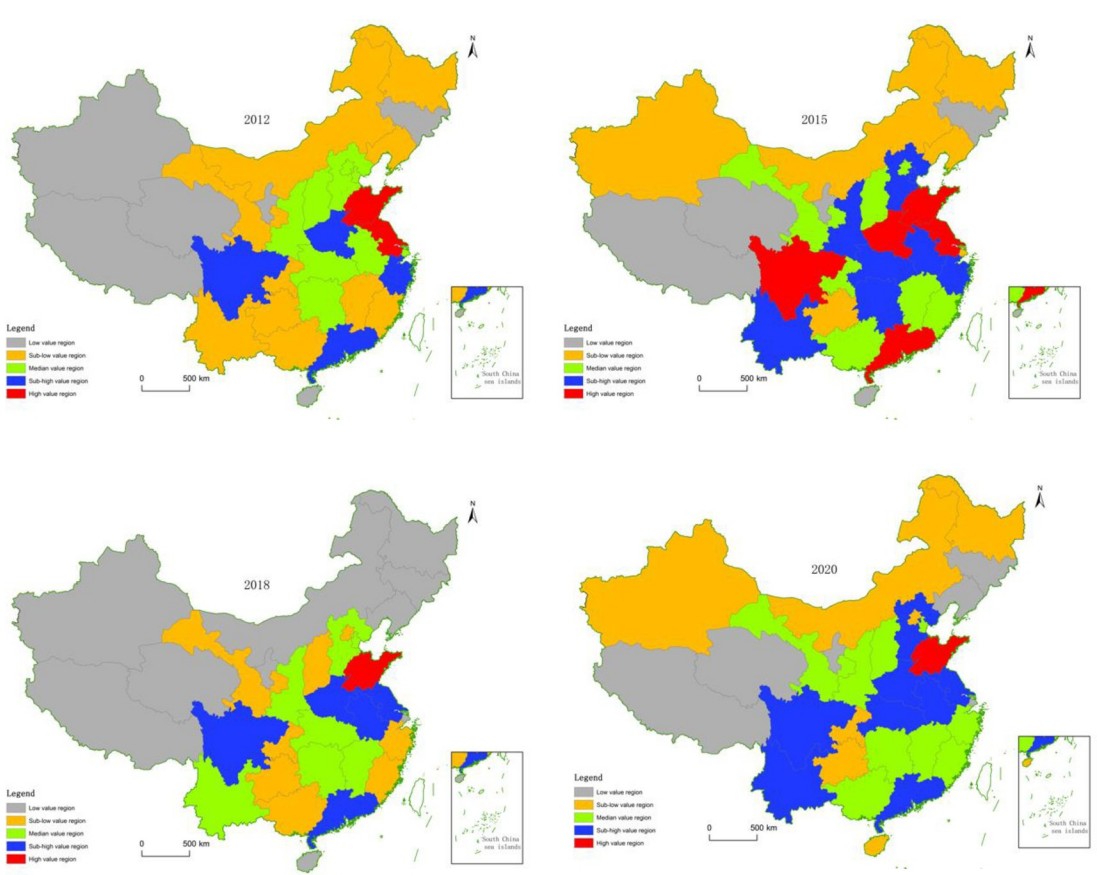

**Fig 1. Evolution of the spatial distribution of pension fund balances.**

the Central region, and Yunnan Province in the Western region. The median area still comes from two contiguous areas and one province. Still, it differs from 2018, namely the contiguous areas of Gansu, Shaanxi, and Shanxi, the contiguous areas of Guangxi, Hunan, Jiangxi, Fujian, and Zhejiang, and include Tianjin. Sub-low and low-value areas are still primarily distributed in the northeast and Western regions. Therefore, the high-value areas are mainly distributed in the Eastern region, and the low-value areas are distributed primarily in the Western and Northeastern regions.

## Theil index for pension fund balances

According to the above division of China's geographical regions and the calculation and decomposition formula of the Theil index, the overall, regional, within-group, and between groups Theil index of China's pension fund balances from 2012 to 2020 can be calculated. As shown in Table 2.

The overall Theil index of pension fund balances has shown a downward trend, from 0.3576 in 2012 to 0.3049 in 2020, which shows that the differences in pension fund balances in various regions are narrowing. The within-group Theil index and the between-group Theil index of pension fund balances also showed a downward trend, falling from 0.2302 and 0.1274 in 2012 to 0.2222 and 0.0828, respectively, and the declines were 0.008 and 0.0446, respectively. The decline in the Theil index is the main reason for the decrease in the overall Theil index.

**Table 2. Theil index decomposition and the contribution rate of pension fund balance.**

| Year | T (Overall) | T$_w$ (Within group) | T$_b$ (Between groups) | Contribution rate (Within group) | Contribution rate (Between groups) | T (Eastern) | T (Central) | T (Western) | T (Northeast) |
|---|---|---|---|---|---|---|---|---|---|
| **2012** | 0.3576 | 0.2302 | 0.1274 | 64.37% | 35.63% | 0.2609 | 0.0491 | 0.3550 | 0.0150 |
| **2013** | 0.3498 | 0.2414 | 0.1085 | 68.99% | 31.01% | 0.2472 | 0.0933 | 0.4072 | 0.0132 |
| **2014** | 0.3110 | 0.2152 | 0.0959 | 69.18% | 30.82% | 0.2608 | 0.0577 | 0.3020 | 0.0125 |
| **2015** | 0.3114 | 0.2155 | 0.0959 | 69.19% | 30.81% | 0.2689 | 0.0559 | 0.2969 | 0.0114 |
| **2016** | 0.3116 | 0.2119 | 0.0997 | 68.02% | 31.98% | 0.2651 | 0.0531 | 0.2914 | 0.0112 |
| **2017** | 0.3070 | 0.2167 | 0.0903 | 70.59% | 29.41% | 0.2808 | 0.0451 | 0.2959 | 0.0061 |
| **2018** | 0.3132 | 0.2201 | 0.0931 | 70.28% | 29.72% | 0.2997 | 0.0444 | 0.2859 | 0.0057 |
| **2019** | 0.3123 | 0.2192 | 0.0931 | 70.18% | 29.82% | 0.3104 | 0.0464 | 0.2724 | 0.0091 |
| **2020** | 0.3049 | 0.2222 | 0.0828 | 72.87% | 27.13% | 0.3037 | 0.0470 | 0.2933 | 0.0100 |

Judging from the contribution of the Theil index of pension fund balances, the contribution of the Theil index within the group is increasing, rising from 64.37% in 2012 to 72.87%. On the contrary, the contribution of the Theil index between groups is decreasing, from 35.63% in 2012 to 27.13% in 2020. This means that although both the within-group Theil index and the between-group Theil index have declined, the within-group Theil index has declined more slowly, and the between-group Theil index has declined faster. Therefore, reducing regional differences in pension fund balances should start with reducing the Theil index within the group.

In addition, judging from the Theil index of each region, the Theil index in the Eastern region is on the rise, rising from 0.2609 in 2012 to 0.3037 in 2020. On the other hand, the Theil indices showed a downward trend in the Central, Western, and Northeastern regions, falling from 0.0491, 0.3550, and 0.0150 in 2012 to 0.0470, 0.2933 and 0.0100, respectively. This further suggests that the decline in the overall Theil index is mainly due to the decrease in the Theil index between the regions and the decline in the Theil index in the northeast, Central, and Western regions.

## Impact factors detection

**Impact factors selection.** The income side of China's pension funds is mainly composed of individual contributions, government financial subsidies, collective subsidies, interest, transfers, and other subsidies for pension insurance. In contrast, the payment side comprises basic and personal account pensions. Based on previous studies, this study considers the availability of data. First, it selects residents' regional GDP and per capita disposable income at the economic level. Second, the working-age population and those aged 65 and above are selected at the population level. Third, fiscal, social security and employment expenditures are chosen at the financial level. Finally, the number of pension fund contributors and the actual number of pension fund recipients are selected at the institutional level. There are eight indicators in total, as shown in Table 3.

## Identification of influencing factors and analysis of strength

Since the geographic detector model is suitable for the analysis where the dependent variable is a numerical variable and the independent variable is a variable type, the "equal percentile" classification method of SPSS 22.0 is used to discretize the selected influencing factor data. First, the independent variables are divided into five categories, and the eight assigned influencing factors are divided into five levels. Then, using the geographic detector model, the balance of

**Table 3. Selection of impact factors.**

| Levels | Impact factors |
|---|---|
| Economic level | Regional GDP (X1) |
| | Residents' per capita disposable income (X2) |
| Population level | The working-age population (X3) |
| | Aged 65 and above (X4) |
| Financial level | Fiscal expenditure (X5) |
| | Social security and employment expenditures (X6) |
| Institutional level | The number of pension fund contributors (X7) |
| | The actual number of pension fund recipients (X8) |

the pension funds in each province from 2012 to 2020 is used as the dependent variable, the impact factor after discretization is used as the independent variable, and the factor detection analysis is performed to obtain the factor influence value q. The collation results are shown in Table 4.

The significance test only residents' per capita disposable income (X2) failed the significance test among the eight indicators, probably because there are significant differences in per capita disposable income in different regions. However, most residents prefer to choose the old-age pension. Moreover, the level of premium payment is the lowest, so although there is a significant gap in the per capita disposable income of residents in different regions, the difference in the payment grades chosen by residents is not substantial.

From the average value of factor strength q, the q value from large to small in 2012–2020 is the number of the working-age population (X3), the number of people over 65 years old (X4), the actual number of pension recipients (X8), and regional GDP (X1), financial expenditure (X5), number of pension contributors (X7), social security and employment expenditure (X6). This shows that, during 2012–2020, the impact factor at the population level is the primary reason for the regional differences in pension fund balances.

From the perspective of the time change, the strength of impact factors at different levels from 2012 to 2020 showed other trends. At the economic level, the q value of regional GDP (X1) dropped from 0.7472 in 2012 to 0.5678 in 2020. At the population level, the q values of the working-age population (X3) and those aged 65 and over (X4) increased first and then decreased, but the overall trend fluctuated. The q value of the working-age population (X3) rose from 0.6752 in 2012 to 0.7575 in 2015 and began declining gradually. By 2020, it had dropped to 0.6968, but it had increased by 0.0216 compared with 2012. The q value of the

**Table 4. Variation trend of factor action strength (q value).**

| Factors | q (2012) | q (2013) | q (2014) | q (2015) | q (2016) | q (2017) | q (2018) | q (2019) | q (2020) | Average |
|---|---|---|---|---|---|---|---|---|---|---|
| X1 | 0.7472*** | 0.7531*** | 0.7336*** | 0.6061*** | 0.5740*** | 0.6247*** | 0.5832*** | 0.5508*** | 0.5687*** | 0.6379 |
| X2 | 0.2818 | 0.2576 | 0.214 | 0.1812 | 0.0885 | 0.0823 | 0.1001 | 0.0979 | 0.0696 | 0.1526 |
| X3 | 0.6752*** | 0.7361*** | 0.7493*** | 0.7575*** | 0.7426*** | 0.7241*** | 0.6957*** | 0.6899*** | 0.6968*** | 0.7186 |
| X4 | 0.6313*** | 0.6782*** | 0.7068*** | 0.7191*** | 0.7213*** | 0.6952*** | 0.6629*** | 0.6474*** | 0.6400*** | 0.6811 |
| X5 | 0.6075*** | 0.6428*** | 0.7165*** | 0.7043*** | 0.6666*** | 0.5872*** | 0.5822*** | 0.5537*** | 0.5493*** | 0.6233 |
| X6 | 0.5768*** | 0.6168*** | 0.4614*** | 0.4627*** | 0.3891*** | 0.4133*** | 0.5140*** | 0.5114*** | 0.4826*** | 0.4920 |
| X7 | 0.3771** | 0.4656*** | 0.5292*** | 0.5602*** | 0.5752*** | 0.5716*** | 0.5323*** | 0.5566*** | 0.5799*** | 0.5267 |
| X8 | 0.5325*** | 0.5997*** | 0.6210*** | 0.6451*** | 0.6497*** | 0.6609*** | 0.6544*** | 0.7030*** | 0.7400*** | 0.6451 |

Note

***, **, and * indicate passing the 1%, 5%, and 10% significance tests, respectively.

population aged 65 and over (X4) rose from 0.6313 in 2012 to 0.7213 in 2016, then began to decline, and dropped to 0.6400 in 2020, but it was 0.0087 higher than that in 2012. At the fiscal level, the q values of fiscal expenditure (X5), social security, and employment expenditure (X6) showed a downward trend in fluctuations, from 0.6075 and 0.5768 in 2012 to 0.5493 0.4826 in 2020, respectively. As a result, their impact on regional differences could be stronger. At the institutional level, the q values of the number of pension contributors (X7) and the actual number of pension recipients (X8) are rising, from 0.3771 and 0.5325 in 2012 to 0.5799 and 0.7400 in 2020, respectively. It is worth mentioning that the q value of the actual number of pensioners has become the most significant factor affecting the balance of pension funds in 2020.

Overall, the impact factors of the economic and financial levels on the regional differences in pension fund balances are decreasing, while the impact factors of the population and system levels on the regional differences in pension fund balances are increasing. In other words, regional differences in pension fund balances are more influenced by demographic and institutional factors.

## Interactive detection analysis of impact factors

One of the significant advantages of geographic detectors is that they can detect the interaction between two influencing factors. This study detects pairwise interaction on eight indicators affecting regional differences in pension fund balances. Table 5 shows the interactive detection results in 2012, 2016, and 2020. It can be found that there is an enhanced relationship between two influencing factors, including two-factor, improved and nonlinear enhanced, and there is no independent or weakened relationship. In 2012, only residents' disposable income (X2) ∩, the number of pension contributors (X7), and residents' per capita disposable income (X2), ∩ the actual number of pension recipients (X8), were nonlinear enhancements (NE). The remaining factor interactions all produced two-factor enhancement (BE). In 2016 and 2020, the range of nonlinear enhancement (NE) produced by factor interaction was expanded to 7 groups. These seven groups are all caused by the interaction between residents' per capita disposable income (X2) and other factors. This shows that although the influence of per capita disposable income of residents (X2) alone fails the p-value test, its interaction with other

**Table 5. Interaction detection results.**

| Factors | 2012 | 2016 | 2020 | Factors | 2012 | 2016 | 2020 |
|---|---|---|---|---|---|---|---|
| X1∩X2 | BE | NE | NE | X3∩X5 | BE | BE | BE |
| X1∩X3 | BE | BE | BE | X3∩X6 | BE | BE | BE |
| X1∩X4 | BE | BE | BE | X3∩X7 | BE | BE | BE |
| X1∩X5 | BE | BE | BE | X3∩X8 | BE | BE | BE |
| X1∩X6 | BE | BE | BE | X4∩X5 | BE | BE | BE |
| X1∩X7 | BE | BE | BE | X4∩X6 | BE | BE | BE |
| X1∩X8 | BE | BE | BE | X4∩X7 | BE | BE | BE |
| X2∩X3 | BE | NE | NE | X4∩X8 | BE | BE | BE |
| X2∩X4 | BE | NE | NE | X5∩X6 | BE | BE | BE |
| X2∩X5 | BE | NE | NE | X5∩X7 | BE | BE | BE |
| X2∩X6 | BE | NE | NE | X5∩X8 | BE | BE | BE |
| X2∩X7 | NE | NE | NE | X6∩X7 | BE | BE | BE |
| X2∩X8 | NE | NE | NE | X6∩X8 | BE | BE | BE |
| X3∩X4 | BE | BE | BE | X7∩X8 | BE | BE | BE |

**Note:** BE means two-factor enhancement, and NE means non-linear enhancement.

factors can be reflected as an enhanced relationship. Therefore, we must attach great importance to the exchange and complementary relationship between various factors; that is, the level of economic development and population structure, fiscal expenditure, and institutional arrangements must develop in a balanced manner and go hand in hand to be more conducive to the sustainability of regional pension funds.

## Discussion

### Theoretical contribution

This study contributes to the theoretical research on pension balances in China in the following ways:

First, while existing research primarily focuses on pension fund investment management methods [42], operating efficiency differences [43], and delayed retirement policy implementation [44], the comparison of pension funds across different countries or regions from the perspective of the growth rate of fund balances has received less attention. This study fills this gap by examining the growth rate of fund balances to compare pension funds. As observed in previous studies, this research confirms that population aging puts considerable strain on pension sustainability [45]. However, it highlights the unique pension pressures faced by different countries or regions, thereby introducing a new comparative dimension to the discussion on the fairness of the pension system.

Secondly, this study takes a comprehensive approach by establishing a framework to analyze the influencing factors that can affect the balance of regional pension funds. It systematically explores the sources of regional gaps in pension funds. While economic and fiscal factors have traditionally been recognized as important contributors to pension fund dynamics, this study uncovers the significant influence of population mobility, particularly the increasing number of international floating populations [46]. This study introduces a fresh perspective on how global population flows impact the development of a country or region's pension industry.

Overall, this study adds to the existing body of knowledge by considering the growth rate of fund balances for comparative analysis and shedding light on the impact of population mobility on pension fund dynamics.

### Practice implication

This study draws several conclusions based on measuring regional disparities in China's pension balance and detecting influencing factors. The regional disparities in China's pension balance primarily stem from economic, population, financial, and system differences. To address and narrow these disparities, actions can be taken at four different levels:

Firstly, at the economic level, the government must prioritize coordinated development. While promoting prosperity in the eastern region, equal attention should be given to accelerating the rise of the northeast, central regions, and the development of the west. Reducing the gap in economic development between regions makes it possible to effectively mitigate the widening trend of regional disparities in pension funds.

Secondly, at the population level, the study highlights the significant impact of population age structure imbalance on regional differences in pension fund balances. To address this, provincial governments should focus on attracting talent to expand the working-age population. Simultaneously, simplifying labor group insurance relationship transfer and continuation procedures can eliminate institutional barriers to labor mobility. Moreover, enhancing the inclusiveness of childbirth policies and promptly implementing relevant policies and supporting measures to encourage childbearing in the central, western, and northeastern regions can

optimize the population's age structure, promote long-term balanced population development, and enhance the financial sustainability of pension funds.

Thirdly, from a financial perspective, the central government should increase fiscal transfer payments to the central, western, and northeastern regions. Finally, at the local level, governments should allocate increased spending towards social security and employment to attract the working-age population.

Lastly, the government should continue promoting and expanding the pension system at the institutional level. This can be achieved by strengthening policy publicity and guidance in regions with smaller pension fund balances, increasing the base of insured persons, encouraging a "pay more, get more" approach, increasing per capita pension payments, and increasing the number of pensioners in the region.

In summary, by addressing the economic, population, financial, and institutional factors, the regional disparities in pension funds can be effectively narrowed, leading to a more equitable and sustainable pension system.

## Limitations

This study has certain limitations. First, considering the data availability, this study's research level is provincial-level macro data. Suppose relevant data at the municipal level can be obtained later. In that case, it can measure the gap in pension fund balances between regions and provinces and analyze the differences between different areas of the same site. Second, because China has unbalanced regional development, it is acceptable and reasonable to have some gaps in pension fund balances. However, this study did not discuss this part. Future research should be more realistic to focus on the unreasonable amount of the regional gap in China's pension fund balance to detect related factors.

## Conclusions

According to the research objectives proposed in this paper, four main conclusions can be drawn from the research results: First, China's regional differences in the growth rate of pension fund balances are pronounced. The growth rate of fund balances in each province is quite different and shows a clear downward trend. Second, pension fund balances show agglomeration characteristics in space, with high-value areas mainly distributed in the Eastern region and low-value areas primarily distributed in the Western and Northeastern regions. Third, when China is divided into four major parts: Eastern, Central, Western, and Northeast, the overall Theil index of pension fund balances shows a downward trend. From the contribution perspective, the contribution of the Theil index within the group is increasing, and that of the Theil index between groups is decreasing. The difference in pension fund balances within the group is the main reason for the regional differences in the overall fund balance. Fourth, the detection results of influencing factors show that the impact of economic and fiscal factors on regional differences in pension fund balances is declining. However, the influencing factors at the population and system levels are increasing in strength on the regional differences in pension fund balances and have become the main factors leading to the regional differences in pension balances. In addition, the interaction detection results show that the superimposed influence of each element is manifested as a two-factor enhancement or nonlinear enhancement relationship.

## Acknowledgments

We would like to sincerely thank the reviewers Simon Grima and Kiran Sood for their invaluable feedback and contributions, which significantly enhanced the quality and rigor of this

paper. Their expertise and insights were instrumental in shaping the final outcome of this research.

## Author Contributions

**Conceptualization:** Songbiao Zhang, Huilin Wang.

**Funding acquisition:** Songbiao Zhang.

**Methodology:** Songbiao Zhang, Xining Wang.

**Project administration:** Huilin Wang.

**Writing – original draft:** Songbiao Zhang, Xining Wang, Huajin Li, Huilin Wang.

**Writing – review & editing:** Songbiao Zhang, Xining Wang, Huajin Li, Huilin Wang.

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
