## [Decision Letter · Decision Letter 0]

11 May 2023

PONE-D-23-10690The geographical pension gap: Understanding the causes of inequality in China's pension fundsPLOS ONE

Dear Dr. Wang,

Thank you for submitting your manuscript to PLOS ONE. After careful consideration, we feel that it has merit but does not fully meet PLOS ONE’s publication criteria as it currently stands. Therefore, we invite you to submit a revised version of the manuscript that addresses the points raised during the review process.

We look forward to receiving your revised manuscript.

Kind regards,

Ercan Özen, PhD

Academic Editor

PLOS ONE

Journal Requirements:

   "This study was supported by the National Science Foundation of China (Project No. 42101172) and the Hunan Provincial Philosophy and Social Science Foundation (Project No. 22YBA133)."

    "This study was supported by the National Science Foundation of China (Project No. 42101172) and the Hunan Provincial Philosophy and Social Science Foundation (Project No. 22YBA133)."

    "This study was supported by the National Science Foundation of China (Project No. 42101172) and the Hunan Provincial Philosophy and Social Science Foundation (Project No. 22YBA133)."

6. We note that Figure 1 in your submission contain map/satellite images which may be copyrighted. All PLOS content is published under the Creative Commons Attribution License (CC BY 4.0), which means that the manuscript, images, and Supporting Information files will be freely available online, and any third party is permitted to access, download, copy, distribute, and use these materials in any way, even commercially, with proper attribution. For these reasons, we cannot publish previously copyrighted maps or satellite images created using proprietary data, such as Google software (Google Maps, Street View, and Earth). For more information, see our copyright guidelines: http://journals.plos.org/plosone/s/licenses-and-copyright.

Reviewers' comments:

Reviewer's Responses to Questions

**Comments to the Author**

1. Is the manuscript technically sound, and do the data support the conclusions?

Reviewer #1: Yes

Reviewer #2: No

2. Has the statistical analysis been performed appropriately and rigorously? 

Reviewer #1: Yes

Reviewer #2: No

3. Have the authors made all data underlying the findings in their manuscript fully available?

Reviewer #1: Yes

Reviewer #2: No

4. Is the manuscript presented in an intelligible fashion and written in standard English?

Reviewer #1: Yes

Reviewer #2: No

5. Review Comments to the Author

Reviewer #1: I am pleased to have the opportunity to review this research paper titled: The geographical pension gap: Understanding the causes of inequality in China's pension funds. This study attempted to explore the the causes of inequality in China's pension funds.

The paper demonstrates an adequate understanding of the relevant literature in the field and cites an appropriate range of literature sources related to the study itself. The paper clearly express its case, measured against the technical language of the field and the expected Knowledge of the journal's readership.

The article flows well and puts the reader into the context of the subject. The aim and objective are well explained and result from the gaps in the subject. The methodology is well explained and enables replication, while the results and conclusions triangulate with the rest of the paper and are well discussed, highlighting some practical implications.

I believe this paper can be published as is and adds value to literature already published in this field.

Reviewer #2: China has seen a sharp reduction of poverty, but also a substantial increase of inequality. As the result of more than two decades of rapid economic growth in China, millions have been lifted out of poverty, resulting in an

impressive decline in the poverty . I will urge authors to write more about literature and focus on the gap.

Kindly elaborate more about research methodology .

Discussion part is not satisfactory.

6. PLOS authors have the option to publish the peer review history of their article (what does this mean?). If published, this will include your full peer review and any attached files.

Reviewer #1: **Yes: **Simon Grima

Reviewer #2: **Yes: **Kiran Sood

---

## [Author Response · Author response to Decision Letter 0]

17 Jun 2023

Response to Reviewer 1 Comments

We extend our heartfelt gratitude for your positive feedback on our manuscript. We fully acknowledge that our work may not have been flawless initially, and we greatly appreciate your suggestions. With your guidance, we have enriched the relevant literature and provided further elaboration on the research methodology employed in our manuscript. Your valuable input has contributed significantly to enhancing the quality and depth of our work. We sincerely appreciate your professional insights and guidance.

Response to Reviewer 2 Comments

Dear reviewer,

We express our heartfelt gratitude to the reviewer for thoroughly examining our manuscript and offering constructive comments that guided our revision process. We have diligently incorporated your valuable suggestions into the manuscript, and we appreciate your time and effort in reviewing our work. Please find detailed responses to your comments and suggestions below. Once again, we extend our sincere thanks to the reviewer for their invaluable contribution.

（1）China has seen a sharp reduction of poverty, but also a substantial increase of inequality. As the result of more than two decades of rapid economic growth in China, millions have been lifted out of poverty, resulting in an impressive decline in the poverty . I will urge authors to write more about literature and focus on the gap.

Thank you for your suggestion. Following your advice, we have incorporated the following content: "Since implementing economic reforms and opening up, China has experienced rapid economic development and substantially reduced absolute poverty. However, the issue of relative poverty remains severe, and the widening income gap poses a significant obstacle to achieving common prosperity (1). Establishing the old-age insurance system has been a crucial policy tool to narrow this income gap. Nevertheless, China faces the unique challenge of “getting old before getting rich” due to the relatively short period during which the old-age security system has been in place and the accelerated population aging, particularly among certain demographic groups (2). 

Initially, China’s pension insurance system was designed to safeguard the well-being of older adults, prevent elderly poverty, and reduce income disparities among older individuals. Currently, the pension insurance system in China has achieved nearly universal coverage among the population, with an increasing participation rate. However, due to inadequate overall planning of the pension insurance fund at the national level, a significant disparity exists in the income and expenditure of pension funds across different provinces. As a result, some provinces face shortfalls in pension fund revenue, resulting in a decline in the growth rate of the fund balance. Consequently, this not only negatively impacts efforts to alleviate poverty among older adults but also exacerbates regional income disparities among this age group, contradicting the original intent of the pension insurance system (3). Furthermore, demographic changes are generating global concerns regarding the long-term financial sustainability of pension schemes (4, 5, 6), a matter of particular urgency for China." 

Furthermore, we have included a comprehensive literature review section, which provides detailed insights within the manuscript.

（2）Kindly elaborate more about research methodology .

Thank you for your suggestion. Following your advice, we have incorporated the following content: “In measuring regional disparities, numerous calculation methods have been utilized in previous studies, including the coefficient of variation, Theil index, standard deviation, and Gini coefficient. However, when evaluating changes in the absolute gap between pension regions, comparing the absolute gaps at different time points can be misleading due to variations in the balance growth rate and the overall balance level. To address this issue, the relative gap is employed in this study, which eliminates the direct impact of the total size and enables a more meaningful assessment of changes in regional disparities over time. In addition, the growth rate of fund balances in each region is calculated to reflect these changes.

Furthermore, China has historically exhibited significant disparities in fund balances among its Eastern, Central, Western, and Northeastern regions, primarily due to their differing geographical locations. The advantage of utilizing the Theil index lies in its ability to measure and compute gaps between and within regions, making it a suitable choice for quantifying the discrepancies in fund balances.

Lastly, this study adopts the geographic detector method to identify multiple influencing factors contributing to regional fund balance gaps. Unlike conventional spatial regression analysis, the geographic detector method does not rely on linear assumptions or conditional restrictions. It offers an objective means of determining the degree of interpretation of independent variables on the dependent variable and provides the added advantage of detecting interactions between these factors.”

（3）Discussion part is not satisfactory.

We appreciate your suggestion. Taking your advice into account, we have made adjustments to the theoretical and practical contributions discussed in the manuscript's Discussion section. These revisions have been thoroughly elaborated upon within the manuscript itself.

---

## [Editor Report · Decision Letter 1]

5 Jul 2023

The geographical pension gap: Understanding the causes of inequality in China's pension funds

PONE-D-23-10690R1

Dear Dr. Huilin Wang,

We’re pleased to inform you that your manuscript has been judged scientifically suitable for publication and will be formally accepted for publication once it meets all outstanding technical requirements.

Kind regards,

Ercan Özen, PhD

Academic Editor

PLOS ONE
---

## [Editor Report · Acceptance letter]

7 Jul 2023

PONE-D-23-10690R1 

The geographical pension gap: Understanding the causes of inequality in China’s pension funds 

Dear Dr. Wang:

I'm pleased to inform you that your manuscript has been deemed suitable for publication in PLOS ONE. Congratulations! Your manuscript is now with our production department. 

Kind regards, 

on behalf of

Dr. Ercan Özen 

Academic Editor

PLOS ONE